# Prognostic Role of Human Leukocyte Antigen Alleles and Cytokine Single-Nucleotide Polymorphisms in Patients with Chronic Myeloid Leukemia Treated with Tyrosine Kinase Inhibitor Drugs

**DOI:** 10.3390/genes15060732

**Published:** 2024-06-02

**Authors:** Samuel Kinde Birru, Ilias Doxiadis, Rawleigh Howe, Tsehayneh Kelemu, Saifu Hailu Chala, Abdulaziz Sherif, Fisihatsion Tadesse, Aster Tsegaye, Amha Gebremedhin, Claudia Lehmann

**Affiliations:** 1José Carreras Research Laboratory, Department of Hematology and Cell Therapy, Faculty of Medicine, University of Leipzig, Johannisallee 32A, 04103 Leipzig, Germany; samuel.kinde@aau.edu.et; 2Armauer Hansen Research Institute, Addis Ababa P.O. Box 1005, Ethiopia; 3College of Health Sciences, Addis Ababa University, Addis Ababa P.O. Box 1176, Ethiopia; 4Laboratory for Transplantation Immunology, University Hospital Leipzig, Johannisallee 32, 04103 Leipzig, Germany; ilidox1@icloud.com; 5Medical Laboratory Scienec, Madda Walabu University, Bale Robe P.O. Box 247, Ethiopia

**Keywords:** cytokine polymorphism, CML, Ethiopian HLA frequencies, tyrosine kinase inhibitor, IL-6, IL-10, TNF-α, IFN-γ, TGF-β, Th17, Treg

## Abstract

Tyrosine kinase inhibitor (TKI) drugs have significantly improved chronic myeloid leukemia (CML) outcomes. Neopeptides from CML cells may induce specific immune responses, which are crucial for deep molecular (DMR) and treatment-free remission (TFR). In this study of Ethiopian patients with CML (n = 162), the HLA alleles and single-nucleotide polymorphisms of five cytokines revealed significant associations with clinical outcomes. Clinically unfavorable outcomes correlated with HLA alleles *A*03:01/02*, *A*23:17:01*, *B*57:01/02/03*, and *HLA-DRB4*01:01* (*p*-value = 0.0347, *p*-value = 0.0285, *p*-value = 0.037, and *p*-value = 0.0127, respectively), while *HLA-DRB4*01:03:01* was associated with favorable outcomes (*p*-value = 0.0058). After assigning values for the ‘low’, ‘intermediate’, and ‘high’ gene expression of the SNPs’ respective cytokine genes, Kaplan–Meier estimates for relapse-free survival, adjusted for age, treatment duration, and relapse risk among patients after the administration of TKIs, indicated that a gene expression ratio above the overall median of TNF-α, IL-6, and the combination of TGF-β1/IL-10, IFNγ, and IL-6/IL-10 TGF-β1 was correlated with a higher likelihood of treatment failure ((RR: 3.01; 95% CI: 1.1–8.3; *p*-value = 0.0261) and (RR: 2.4; 95% CI: 1.1–5.2; *p*-value = 0.022), respectively). Multi-SNPs, surpassing single-SNPs, and HLA allele polymorphisms showed promise in predicting outcomes of patients with CML during TKI treatment, prompting further exploration into their potential utility.

## 1. Introduction

The distinguishing feature of chronic myeloid leukemia (CML) is the presence of the Philadelphia chromosome (Ph), detected in over 95% of patients. This genetic mutation involves the reciprocal exchange of the lengthy segments of chromosome 22 with the BCR gene and those of chromosome 9 with the ABL gene (9;22). This translocation is responsible for the persistent activation of tyrosine kinase activity and the suppression of apoptosis [1,2]. Tyrosine kinase inhibitor (TKI) drugs have revolutionized CML treatment, extending the lives of patients in the chronic phase (CP) to near-normal expectancy, and for some individuals in deep molecular remission (DMR), they can be considered for a safe treatment-free remission (TFR) [3,4,5]. Nonetheless, up to 40–60% of patients can relapse following the discontinuation of TKI treatment [6] despite being in the CP during the course of TKI treatment. Both innate [7] and adaptive immunity, such as reduced immune suppressive cells [8,9] and mesenchymal stromal cells with immunomodulatory roles [10], can impact DMR and thereby TFR.

Peptides from the BCR-ABL breakpoint can be processed within the cytosol, loaded onto major histocompatibility complex (MHC) molecules, and presented on the surface of CML cells, potentially allowing T cell recognition [11]. This suggests that ‘neoantigens’ derived from the BCR-ABL fusion gene may provoke specific immune responses, making CML cells a potential target for the immune system [12]. The MHC [13] comprises a series of genes and their products involved in protecting the respective individual against viral or bacterial infections and parasites. It is the most polymorphic system in humans. The products are named human leucocyte antigens (HLAs). The variability is specific and differs from landmark to landmark, reflecting the provenience of the individual. Functionally, HLA antigens are the counterparts of the T, B, and NK cell receptors and are involved in the self-non-self-recognition [14]. HLA is used diagnostically in stem cell and solid organ transplantation. An analysis of HLA ligands in samples of patients with primary CML has revealed previously unknown CML-associated antigens capable of triggering diverse T cell responses [15]. Additionally, the efficacy of TKIs is influenced by increased NK cell levels, which rely on interactions between NK cell receptors (KIRs) and HLA alleles [16]. A study of the European Group for Blood and Marrow Transplantation (EBMT) indicated that the presence of HLA-B8, particularly when co-expressed with HLA-A3, and the sole expression of HLA-DR4 were associated with a diminished incidence of CML [17,18]. However, these findings necessitate further assessment using cutting-edge sequencing technologies across diverse patient groups and are especially confirmed in other populations. 

In Iranian patients, haplotypes such as *HLA-DRB1*11/-DQB1*03:01/-DQA1*05:05* and *HLA-DRB1*04/-DQB1*03:02/-DQA1*03:01* seem to be predisposed to CML [19,20]. In contrast, in Turkish patients, there was a protective effect of *DRB1*11* and *B*35*. Here, DRB4 homozygosity and *B*37* were regarded as risk factors [21]. The validation of genetic markers is a prerequisite for a reliable prognostic tool in managing CML and achieving TFR. Specific allele significance requires confirmation across diverse populations of patients with CML.

The level and activity of cytokines is attributed to single-nucleotide polymorphisms (SNPs) within the promoter region or other regulatory sequences of cytokine genes, which in turn impact the peripheral levels of cytokines and thus pathogenesis [22,23,24]. Population studies have indicated a strong association of SNPs with primary immune thrombocytopenia among Caucasian adults [25], the risk of leukemogenesis in a non-Chinese population [26], the severity of autoimmune hemolytic anemias in an Italian population [27], and the risk and overall survival of diffuse large B cell lymphoma in an Arab population [28]. These reports emphasize the importance of cytokine gene SNPs across diverse populations, shedding light on the intricacies of homeostasis and disease development as well as HLA allele polymorphism.

Functionally, TGF-β1, in conjunction with IL-6 or IL-21, supports Th17 cell differentiation. However, specific SNPs at positions +869 (T/C) and +915 (G/C) in the first exon of the TGF-β1 gene alter codon 10 (from leucine to proline) and codon 25 (from arginine to proline), resulting in varied expression levels [29]. IL-10, also known for its pivotal role in enhancing tumor-specific immune surveillance and curbing detrimental inflammation, is increasingly recognized in the host’s battle against malignant cells. Three crucial SNPs in its promoter region (−1082, −819, and −592) influence IL-10 production in cultured cells, particularly when an ‘A’ is present at positions −1082 and −592, resulting in decreased IL-10 levels [30,31], which might curb deleterious inflammatory responses. 

Similarly, functional studies have revealed that the G allele of the SNP variant −764G/C, situated nearby the IFN-γ promoter region, enhances promoter activity by a factor of two to three fold, correlating with both viral clearance and better treatment response in individuals with HCV infection [32]. In the context of malignancies, TNFα SNP (−863C/A, −857C/T, −308 G/A, and −238 G/A) was strongly associated with a higher risk of hepatocellular carcinoma (HCC) [33], and the different combinations of SNPs of TNFα (−238G/A or −308G/A) with IL-10 (−592C/A) were highly prevalent among patients with cervical cancer. PBMC (samples from patients with cervical cancer) activated with PHA also indicated that the TNFα (−308A/A) genotype demonstrated increased proliferation rates, elevated IL-4 and TGF-β, and decreased IL-2 levels [34], and it was postulated that SNPs of cytokine genes act as the potential predictors of cervical HPV infection progression to neoplasia. 

In CML, cytokine gene polymorphism other than histocompatibility disparity had a vivid impact on the incidence of patients with CML compared to the healthy control group. The most frequent genotype/haplotype among patients with CML was TGF-β1TG/TG (at codon 10G/G and codon 25G/G), which correlates with high gene expression, followed by IFN-γA/T (at intron 1, +874A/T), which correlates with intermediate gene expression, and IL-10 ACC/ACC (−1082A/A, −819C/C, −592C/C) and ATA/ATA (−1082A/A, −819T/T, −592A/A), both of which correlate with low gene expression. In contrast, there is a low frequency among CML vs. control of the genotypes TGF-β1CG/CG (codon 10C/C and codon 25G/G) and IL-10 GCC/ACC (−1082G/A, −819C/C, −592C/C), both of which correlated with intermediate gene expression [35]. However, according to the study, the production of TGF-β1 in patients with CML is higher and IL-10 is lower than in normal subjects, suggesting the possibility of immune bias during the immunological surveillance of CML cells.

In addition to cytokine SNPs, another pivotal factor in immunogenetics is the polymorphism of human leukocyte antigens (HLAs). 

Studies on HLA frequencies for the African population are scarce, with limited and outdated references, particularly for the Ethiopian population, often relying on now obsolete methods [36,37,38]. Only a few African studies, such as Abshir et al., reported high-resolution HLA data from Somalia [39]. Most genetic CML studies were attributed to the non-African population. Immune–genetic studies among patients from African populations have been crucial for understanding diverse genetic variations. These studies have provided insights into CML disease progression, TKI response, and possible genetic adaptations. The SNP cytokine analysis was conducted based on arbitrary values assigned for ‘low’, ‘intermediate’, and ‘high’ gene expression, respectively. This approach allows us to simulate and compute the ratios of the cumulative sum of five cytokines (IL-10, IL-6, IFN-γ, TGF-β1, and TNFα) and to calculate the Kaplan–Meier failure estimate of patients with CML with favorable and unfavorable treatment outcomes in the course of treatment for up to 12 years.

Histocompatibility disparity was observed among the patients regarding their treatment outcomes. In addition, a couple of new alleles were detected, which are still under investigation and require confirmation. One advantage of this study was that the HLA genotyping was performed using next-generation sequencing (NGS) with three-field resolution, which could be an asset for future population genetic studies.

## 2. Materials and Methods

### 2.1. Cohorts

A total of 162 and 63 cytogenetically confirmed Ph+ chronic myeloid leukemia (CML) patients were enrolled consecutively. Those patients were visiting the hematology clinic at Tikur Anbessa Specialized Hospital (TASH), Addis Ababa University (AAU), and Addis Ababa, Ethiopia. The study was approved by the Institutional Review Board (IRB), CHS-IRB 015/2023, of the college of health sciences of the AAU. Written informed consent was received from all participants. The patient cohort was divided into two subgroups, favorable and unfavorable, in terms of TKI response according to the updated NCCN guidelines for the treatment of CML [40]. 

In the overall cohort of patients with CML, 63/162 individuals (41.7%) underwent random selection for further assay—cytokine single-nucleotide polymorphism (SNP) analysis. Within this subset, the pre-treatment median age and duration were 37 years (range: 19–69) and 48 months (range: 1–144), respectively, with 69.8% exhibiting a high Sokal [41] risk score (Table 1).

During data analysis, subjects in accelerated or blast crises and those solely on cytoreductive therapy or those who were TKI naïve during treatment were excluded. Among 51 patients in the chronic phase of CML undergoing TKI treatment (81%), either as a first-line or subsequent therapy, 29 (46%) experienced unfavorable clinical outcomes, while 22 (35%) experienced favorable ones. The median (range) of BCR-ABL1 (IS) was 42.2 (2.0–78.2) for unfavorable outcomes and 0.02 (0.0001–0.25) for favorable ones (Table 2).

### 2.2. DNA Preparation

DNA was isolated from EDTA blood samples according to the manufacturer’s recommendation (QIAamp DNA Blood Mini Kit, QIAGEN, and Hilden, Germany). In short, Qiagen columns (QIAamp DNA mini kit; Qiagen, Hilden, Germany) were used for the extraction of DNA from total white blood cells obtained from EDTA-treated peripheral venous blood collected with Vacutainer Tubes (Becton Dickinson). Briefly, samples were mixed with four-fold ice-cold NH_4_Cl lysing buffer (pH 7.4), kept for 20–25 min on ice until RBC lysis was apparently complete. After washing with ice-cold PBS, pelleted total white blood cells were lysed in guanidine thiocyanate buffer (GTC) containing 8 µL/mL β-mercaptoethanol. A QIAshredder spin columns (QIAGEN #79656) were used to extract DNA according to the manufacturer’s protocol. The purity and quantity of eluted total DNA was measured using a Thermo Scientific™ NanoDrop™ One Spectrophotometer (Waltham, MA, USA).

### 2.3. Cytokine Genotyping

Cytokine genotyping was performed by the polymerase chain reaction-sequence-specific primer (PCR-SSP) method, using the Cytokine Genotyping Tray kit according to the manufacturer’s instructions (Cytokine Genotyping Tray, One Lambda Canoga Park, CA, USA). Briefly, the amplified products of TNF-α, TGF-β, IL-10, IL-6, and IFN-γ alleles and internal control, the human β-globin gene, were analyzed via agarose gel electrophoresis. The interpretation of SNPs (see Table 3) for these five cytokines was based on the presence or the absence of specific amplified fragments.

Using simple descriptive statistics, patients with and without a favorable clinical outcome were compared based on the respective SNP that were associated phenotypically with low-, intermediate-, or high-gene expression levels of each cytokine. Furthermore, the ratio of pro-inflammatory regulatory cytokine (IFN-γ, TNF-α, IL-6) to anti-inflammatory regulatory cytokine (IL-10, TGF-β1) gene expression level was computed among patients with CML having favorable and unfavorable clinical outcomes.

### 2.4. HLA High-Resolution Typing

HLA typing was performed using commercial test kits, the AllType NGS 11-Loci (One Lambda, West Hills, CA, USA) and AlloSeq^®^ Tx 17 (CareDx, San Francisco, CA, USA). For some samples, we additionally used the NANOTYPE^TM^ (Omixon Biocomputing Kft., Budapest, Hungary) system to clarify the results. The sequencing was carried out on a MiSeq Sequencing device (Illumina, San Diego, CA, USA), strictly following the manufacturer’s recommendations. Furthermore, samples were confirmed with the NANOTYPE^TM^ using the Nanopore technology (Oxford Nanopore, Oxford, UK), especially in the case of rare and new alleles. The laboratory in Leipzig (Germany) has European (EFI) and national (DAkkS) accreditation for NGS HLA typing. 

### 2.5. Data Analysis and Interpretation

The cytokine data obtained through PCR-SSP was entered into an evaluation form provided by the company, and SNP-associated cytokine expression levels (‘low’, ‘intermediate’, or ‘high’) were assigned following the provided protocol. To facilitate statistical analysis, arbitrary values (1, 2, or 3, respectively) were assigned to each level of cytokine expression. To evaluate the immunological homeostasis of patients with CMOL based on SNP-related cytokine expression levels, the sum ratio of multiple cytokines—both pro-inflammatory and anti-inflammatory—in various combinations was computed. The overall median of all patients served as a cutoff for Kaplan–Meier analysis (Mantel–Haenszel model) to estimate the probability of relapse-free survival in the favorable versus unfavorable groups (refer to the analysis examples shown below). The analysis was adjusted for variables such as age and treatment duration (Stata Statistical Software, version 13, StataCorp LLC, College Station, TX, USA). The primary objective of this statistical approach was to evaluate how subject’s cytokine SNP affects the risk of relapse.

The calculation of the ratio of cytokine cocktail: $$\frac{\{\mathrm{TNF}\alpha +\mathrm{IL}6+\mathrm{TGF}\beta 1\}}{\{\mathrm{IL}10+\;\mathrm{IFN}\gamma \}}=\frac{\left\{(G/G‘low’=1)+(G/G‘high’=3)+(T/TG/G‘high’=3)\right\}}{\left\{(GCC/GCC‘high’=3)+(T/A‘Intermediate’=2)\right\}}=7/5=1.4$$

If the overall (n = 51) median {(n + 1) ÷ 2}th ratio of the cytokine cocktail was 1.0, then, the cutoff for Kaplan–Meier would be set at a value of ‘1’, and the probability of treatment failures of the above subject (1.4) was calculated using K-M. 

The HLA results were determined by analyzing the fastq files in the Type Stream Visual (TSV) software v2.0 (One Lambda) with the AlloSeq Assign v1.02 software (CareDx) or with NanoTYPER^TM^ v1.1.0 (Omixon). The current IMGT/HLA database at the time of the analysis was used for the HLA analysis [42]. For statistical analysis, we used SPSS v29.

In the present study, we directly compared the HLA typing results of the favorable and unfavorable cohorts making use of the one-tailed Fisher exact test, without making use of a control cohort, since the patients were recruited from all Ethiopian regions.

## 3. Results

### 3.1. Summary and Descriptive Statistics of the SNP vs. Clinical Outcomes

For the initial analysis of the cytokine polymorphism of all patients with CML on TKI treatment (n = 51), we observed that some cytokine polymorphisms do not occur in this Ethiopian patient cohort, such as TGF-β T/T C/C (low), TGF-β T/C C/C (low), and TGF-β T/T G/C (intermediate) (Table 3). 

In the total cohort of patients with CML (Table 3), 60 out of the total 63 cases of patients with CML (95%) had TNF-α polymorphism associated with ‘low’ gene expression (G/G), and the proportion among subgroups was similar. Among the subset of patients with CML in the chronic phase undergoing TKI initiation (n = 51), ‘intermediate’ TGF-β1 gene expression was 4/22 (18%) for patients with favorable and 9/29 (31%) with unfavorable clinical outcomes. Within the favorable subgroup, G/G ‘high’ IL-6 gene expression was evident in 19/22 individuals (86%), whereas the unfavorable group exhibited a lower expression in 20/29 cases (69%). The IFN-γ A/A ‘low’ gene expression was 9/22 (41%) and 18/29 (62%) among favorable and unfavorable clinical outcomes, respectively. For IL-10, the GCC/GCC ‘high’ gene expression among favorable and unfavorable subgroups were 8/22 (36%) and 15/22 (21%), respectively. Conversely, in the unfavorable subgroup, 15/29 individuals (52%) demonstrated a slightly higher proportion of ‘intermediate’ gene expression levels for IL-10 with GCC/ACC and GCC/ATA compared to those in the favorable subgroup, i.e., 8/22 (36%). 

### 3.2. Estimates of the Probability of Relapse-Free Survival

As per Table 3, no statistically significant association was observed between clinical outcomes and the individual cytokine gene expression profiles. However, with the simultaneous computation of multiple cytokines, the following pattern was unveiled.

Figure 1 depicts the median ratio of multiple cytokine gene expression profiles pertinent to designated SNPs, plotting the treatment duration and success and failure variables as remission and relapse, respectively. The relative risk (RR), 95% CI, and *p*-values were calculated using Kaplan–Meier based on the Mantel–Haenszel model (Figure 1a–d). 

The ratio of the combination of TNFα+IL6+TGFβ1/IL10+IFNγ gene polymorphism was computed, and a possible association with the probability of relapse-free survival was explored. Kaplan–Meier (Mantel–Haenszel model) estimates of the probability of relapse-free survival, for a gene expression ratio above the overall median, were relatively high (RR: 3.01; 95% CI: 1.1–8.3; *p*-value = 0.0261) (Figure 1a). The ratio of the results of IL6/IL10+TGFβ1 was computed. Similarly, subjects with a gene expression ratio above the overall median were 2.4 times more at risk for relapse during the period of TKI treatment (RR: 2.4; 95% CI: 1.1–5.2; *p*-value = 0.022) (Figure 1b). The effect of individual cytokine SNPs (ratio of IL6/IL10) was not statistically significant (RR: 0.445; 95% CI: 0.180–1.1; *p*-value = 0.072) nor was the ratio of IL6/TGFβ1 (RR: 0.013). Though patients with a higher than median gene expression (ratio of TNFα/IL10) had twice the risk for relapse, it was not statistically significant (RR: 1.92; 95% CI: 0.877–4.20; *p*-value: = 0.097). Regarding the ratio of IL6+TNFα/IL10, the relative risk of relapse is twice as high among patients with a higher than median gene expression profile (RR: 2.05; 95% CI: 0.94–4.46; *p*-value = 0.064), albeit not significant (Figure 1c). The ratio of IFNγ+TNFα/IL10, in patients with a gene expression profile above the median ratio, had twice the risk of relapse during the treatment period of 10 years or less (RR: 1.94; 95% CI: 0.89–4.23; *p*-value = 0.0876) (Figure 1d).

### 3.3. HLA Polymorphism in the Cohort of Patients Treated for CML

Among all patients with CML in this study, we observed a high frequency of *A*02:01:01, A*03:01:01,* and *A*68:02:01* for A locus alleles, ranging from 8.3 to 16.0%. In the B locus, the most frequently observed alleles were the following: *B*07:02:01, B*15:220:01, B*49:01:01*, and *B*53:01:01* in the range of 7.4% to 9.6%. For the C locus, the alleles *C*04:01:01, C*07:01:01*, and *C*06:02:01* were most prevalent (13.6–14.2%). Interestingly, the allele *B*15:220:01* was found with a frequency of 7.7% in the Ethiopian population, much higher than the observed 1.2% frequency in Gaza, 0.2% in Brazil, and <0.02% in Europe [43]. According to Schell et al. [44], this B allele is regarded as a conserved haplotype associated with *A*66:01*, *C*12:03*, and *DRB1*07:01*. Many of the alleles observed at a high prevalence in this study have also been reported in other populations. These include *A*02:01:01, A*03:01:01, B*07:02:01*, and *C*07:01:01*, as well as *HLA-A*30, B*53*, *and B*58*, which are common in African populations (Table 4).

For HLA class II (DRB1, DRB345, DQA1, DQB1, DPA1, DPB1), we observed the DRB1 alleles *DRB1*07:01:01*, *DRB1*13:02:01*, *DRB1*01:02:01*, *DRB4*01:03:01, DRB3*03:01:01,* and *DRB3*02:02:01* with frequencies greater than 10%. The alleles of the *DRB1*12* group are missing in the studied cohort. With respect to the DQA1 locus, the most common alleles in this Ethiopian CML cohort were *DQA1*01:02:01*, *DQA1*02:01:01*, *DQA1*01:01:02*, and *DQA1*05:01:01* (each >10%). For the DQB1 locus, the three most frequent alleles were *DQB1*02:02:01*, *DQB1*05:01:01*, and *DQB1*06:04:01*. For DPA1, we found *DPA1*01:03:01* (>56%) followed by *DPA1*02:01:01* (>24%). Finally, for the DPB1 locus, the most frequent alleles were *DPB1*02:01:02* (>23%), *DPB1*04:01:01* (>20%), and *DPB1*17:01:01* (>10%). Most of these HLA class II alleles were also found in other populations (Table 5), including European Caucasian, such as *DRB1*15:03:01*, *DQB1*06:09:01*, *DPB1*15:01:01*, and *DPB1*17:01:01* [43].

In the treated cohorts, we found less HLA-*A*23:17:01* (*p*-value = 0.0285) in the favorable group compared to the unfavorable group. The frequency of *A*03* is increased by a factor of 2.3 in the unfavorable group (*p*-value = 0.0347). *B*57:03:01* showed a higher frequency of 7.3% in the favorable group compared to the unfavorable group (2.4%) (*p* = 0.0451). A comparison of allele frequencies of *HLA-DRB4*01:01:01* and *HLA-DRB4*01:03:01* revealed a significant accumulation of the first in the unfavorable group (9.2%), whereas the second was found to be increased in the favorable group with a frequency of 29.2% (*p* = 0.0127 and 0.0058, respectively). For all other HLA loci (C, DRB1, DQA1, DQB1, DPA1 and DPB1), no significant differences between the two cohorts could be found (Table 4).

The HLA polymorphism in the Ethiopian cohort with CML is shown in Table 4, reflecting the expected significant differences in the European and other cohorts (Table 5). Several possible new alleles were observed during this study and are under investigation [45].

**Table 5 genes-15-00732-t005:** HLA class I and HLA class II allele frequencies in various populations.

Population	Region	Most Common HLA Alleles	Remarks	References
Europe	Romania	A*02:01, A*03:01, A*01:01, A*24:02, A*26:01, B*51:01, B*08:01, B*35:01, B*14:02, B*07:02, DRB1*03:01, DRB1*16:01, DRB1*11:01, DRB1*07:01, DRB1*15:01	donors	[46]
	Germany	A*02:01, B*07:02, C*07:01, DRB1*15:01, DRB1*07:01, DQB1*03:01, DQB1*06:02	healthy, ill people, COVID-19 infected	[47]
	Europe	A*01:01, A*24:02, A*33:01, A*68:01, B*08:01, B*44:01, B*57:01, C*12:02, DRB1*01:01, DRB1*01:02, DRB1*03:01, DRB1*04:01, DRB1*04:03, DRB1*07:01, DRB1*11:01, DRB1*11:04, DRB1*13:01, DRB1*13:02, DRB1*15:01, DQB1*02:01, DQB1*02:02, DQB1*03:01, DQB1*03:02, DQB1*04:02, DQB1*05:01, DQB1*05:02, DQB1*05:03, DQB1*06:01, DQB1*06:03, DQB1*06:04, DPB1*01:01, DPB1*02:01, DPB1*04:01, DPB1*04:02, DPB1*06:01, DPB1*11:01		[48]
	European descent group	A*02 (22.4%), A*03 (15.7%), A*24 (9.4%), B*07 (14.6%), B*44 (10,6%), B*51 (7.3%), C*07 (24.6%), C*03 (12.6%), C*06 (11.5%), DRB1*15 (15.2%), DRB1*04 (14.3%), DRB1*03 (11.5%)	all	[49]
Africa	Sudan, Chad, Burkina-Faso	B*53	no A*25, A*26, low A*11, no B*48, B*54, B*67	[50]
	Ethiopia	DR3, DR4	low DR5, DR8, DR10	[37]
	Kenya, Tanzania, Uganda	A*02:01:01 (10.90%), B*58:02 (8.79%), C*06:02:01 (16.98%), DPA1*01:03:01 (40.60%), DPB1*01:01:01 (23.45%), DQA1*01:02:01 (31.03%), DQB1*03:01:01 (21.79%), DRB1*11:01:02 (11.65%), DRB3*02:02:01 (31.65%), DRB4*01:01:01 (10.50%), DRB5*01:01:01 (10.50%)	higher homozygosity: HLA-B, DQA1, DQB1, DRB1 high frequency: DRB1*11 (21.51%)	[51]
	Gabonese, Liberians	DR15/16 DQB1*06:02	Lebanese low DR11	[38]
	African descent group	A*30 (13.1%), A*68 (12.8%), A*02 (12.3%), B*15 (12.6%), B*58 (11.7%), B*44 (9.9), C*07 (16.8%), C*06 (16.4%), C*04 (12.8%), DRB1*03 (17.2%), DRB1*13 (16.1%), DRB1*11 (16%)	all	[49]
Asia	Republic of Korea	A*33:03:01-B*44:03:01-C*14:03-DRB1*13:02:01-DQB1*06:04:01-DPB1*04:01:01 (2.89%), A*33:03:01-B*44:03:01-C*14:03 (4.91%), DRB1*08:03:02-DQA1*01:03:01-DQB1*06:01:01-DPA1*02:02:02-DPB1*05:01:01 (5.41%), DRB1*04:05:01-DRB4*01:03:01 (12.72%), DQA1*01:03:01-DQB1*06:01:01 (13.01%), DPA1*02:02:02-DPB1*05:01:01 (30.83%)	adults from medical school	[52]
Arab Emirates	Saudi	A*02:01:01G (20.2%), A*24:02:01G (7.5%), B*51:01:01G (19.0%), B*50:01:01G (12.3%), C*06:02:01G (16.7%), C*07:02:01G (12.2%), DRB1*07:01:01 (15.7%), DRB1∗03:01:01G (13.3%); DQB1∗02:01:01G (29.9%), DQB1∗03:02:01G (13.2%); andDPB1∗04:01:01G (35.2%), DPB1∗02:01:02G (21.8%)	stem cell donors	[53]

## 4. Discussion

### 4.1. Cytokine Polymorphism

Unfortunately, due to a lack of healthy cohorts as controls, we were limited in making strong inferences, being a general limitation of this study. Therefore, we made comparisons between the favorable and unfavorable groups of patients treated with TKIs. The immunomodulatory effect of TKIs and the achievement of durable DMR, which is a prognostic condition for TFR, and the presence of quiescent TKI-insensitive residual CML leukemic stem cells (LSCs) during sustained TFR draw significant scientific attention in comprehending the immune landscape of patients with CML, which could perhaps underscore the pivotal role of effector and suppressor arms of the immune landscape [54,55,56,57]. 

Our study did not probe the immune cellular compartments, but we rather focused on the genetic landscape of cytokine gene SNPs and HLA allele polymorphisms, which could be a detrimental factor in the balance of effector and suppressor polarization of the immune system. Hence, we examined the concerted effect of multiple cytokine SNPs and HLAs and their correlation with the prognosis of patients with CML.

Clinical outcomes were categorized into two types, favorable and unfavorable, based on the European Leukemia Net (ELN) guidelines, in reference to the treatment milestone achieved during treatment. 

Overall, patient cohorts with unfavorable clinical outcomes, in the absence of TK domain mutation, had poor hematological remission: WBC (median: 19.2 Range: 2.8–180) vs. patients with favorable treatment outcome (median: 6.05; range: 3.4–10.7) (Table 2). These are not particularly surprising clinical phenomena: poor hematological remission has previously been associated with an immunosuppressive trait due to the heightened expression of the PD-L1 gene in monocytes and basophils, as well as other genes linked with reactive oxygen species (ROS) in neutrophils [54,55], thus supporting the evidence for the ELN advice of pre-treating high leukocyte counts with cytoreductive drugs prior to TKI initiation [58]. 

In view of individual polymorphisms and the cocktails of five cytokine SNPs and 11 loci HLA polymorphisms, an appraisal of the immunomodulatory activity of TKI on overall clinical outcomes is noteworthy.

In general, IL-6 has been associated with a wide variety of inflammatory states [56]. Thus, it became the cytokine grabbing perhaps the most research attention during the COVID-19 pandemic.

In our patients, G/G or G/C—both ‘high expressors’ for rs1474348, IL-6 (−174)—were predominant in all cohorts, which was in line with the high allele frequency in an African population [57]. Of the two, the frequency of the G/G genotype was 86% and 69% among the favorable and unfavorable subgroups, respectively, and the G/C genotype was 14.0% and 28.0%, respectively.

In Turkish patients with CML vs. healthy controls, there was a significant difference in the frequency of the genotype: G/G (88.3% vs. 50% *p*-value = 0.010) and G/C (12% vs. 43.2%, *p*-value = 0.002) [22]. Even though our study lacks healthy controls, the proportion of G/G was higher than G/C in patients with CML with both favorable and unfavorable outcomes, which is in line with the Turkish study. With regard to functional characteristic, G/C and G/G genotypes (both associated with ‘high’ gene expression of IL-6) were significantly associated with the pathogenesis of Parkinson’s disease [59] and reduced disease-free survival of breast carcinoma [60], respectively. The C/C genotype (associated with ‘low’ expression of IL-6) appears to benefit the immune landscape, possibly by avoiding a vigorous inflammatory response.

The benefit of low expression of IL-6 could be partially explained by the study finding that DC-mediated T cell activation was enhanced in IL-6 KO (knock out) mice [61], and the level of serum IL-6 levels in CML-blast crises was higher than that in the chronic phase, and the level in the chronic phase was higher than that in normal controls [62]. 

The polymorphism of the TNF-α (−857) C/T TT genotype ‘higher expressers’ was associated with the risk of AML leukemogenesis [63]. Other studies focusing on the TNF-α rs1800629 SNP (−308G/A) have revealed that the G/G ‘low expresser’ genotype was associated with a high risk of adult B-ALL in Sudanese patients, while the −308G/A polymorphism was linked to SLE [64,65] and AML mortality [66]. In our study, 48 out of 51 patients (94.1%) exhibited the G/G genotype. Similar findings of a lack of a significant prognostic role among patients with CML were reported elsewhere [35].

IFN-γ crucially combats viruses and exhibits anti-tumor and anti-proliferative properties [67]. Nuclear factor kappa-B (NF-κB) preferentially binds to +874T allele, perhaps boosting IFN-γ expression and affecting the pathogenesis of tuberculosis [68], and T carriers had statistically increased CML risk in a dominant model (i.e., TT + TA vs. AA) [69]. With regard to its functional impact, IFN-γ stimulation resulted in the upregulation of the major histocompatibility complex (MHC)-II and its master regulator class II transactivator (CIITA) in CML stem/progenitor cells, and the enhanced proliferation of the responder CD4^+^CD69^+^ T cells [70]. The proportion of T/T genotypes (high expressers) was very low in all patients with CML, 4/51 (7.8%). Though difficult to make a direct inference, due to the lack of healthy cohorts as a control, low gene expression could have a possibly negative link with CML, which prompts the necessity for further study. 

Regarding the SNP TGF-β1 (codons 10 and 25), there was no significant (*p*-value > 0.05) difference between patients with CML with favorable and unfavorable outcomes. From the overall cohorts, only 2/22 and 1/29 had ‘low expressers’ among favorable and unfavorable clinical outcomes. A clinically study [71] indicated that poor TKI responders had an enrichment of genes associated with the TGF-β and TNF-α pathways and a quiescence signature that maintains leukemia-initiating cells in chronic myeloid leukemia [72,73]. 

IL-10 functions at the crossroads of immune stimulation and suppression in different malignancies, according to a meta-analysis [31]. Three crucial SNPs in its promoter region (−1082, −819, and −592) influence IL-10 production, and when an ‘A’ is present at positions −1082 and −592, this results in decreased IL-10 levels [30]. In our study, a high frequency, namely, 9/14 (64.3%), of ‘high-expression’ IL-10 genotypes (GCC/GCC; −1082G/G, −819 C/C, −592 C/C) was among the favorable clinical outcomes, while ‘intermediate-expression’ IL-10 (GCC/ACC, ATA; −1082G/G, −819 C/C, T, −592 C/C, A) was higher (15/22 (68.2%)) among unfavorable clinical outcomes (Pearson χ^2^ =3.6526; *p*-value = 0.056). The high expression of IL-10 appeared to contribute to a favorable clinical outcome. In line with this, findings from the ‘KIWI’ study suggested that the TKI withdrawal syndrome (TWS) could be linked to a decrease in IL-10 levels [71]. Nevertheless, an earlier work suggested that CML mesenchymal stem cell (MSC) overexpressing immunomodulatory factors TGFβ1, IL-6, and IL-10 might contribute to CML immune escape through the activation of myeloid-derived suppressor cells (G-MDSC) [69,74]. Based on Figure 1a,b, the gene expression ratio of TNFα+IL-6+TGFβ1/IL-10+IFNγ and IL-6/IL-10+TGFβ1 above the overall median indicates a high probability of relapse, namely, a factor of 3 and 2.4 times, respectively. These intricate relationships may simply highlight the necessity for multi-cytokine computation rather than relying on a single level of cytokine as a potential prognostic tool. 

The ratios of IL-6/IL-10 and TNFα/IL-10 showed no statistically significant differences among patient groups (RR: 0.445, 95% CI: 0.180–1.1, *p*-value = 0.072; and RR: 1.92, 95% CI: 0.877–4.20, *p*-value = 0.097, respectively). This clearly shows that a single or double cytokine would not provide a strong prognostic value. 

In another combination (Figure 1c,d), the ratio of IL-6+TNFα/IL-10 and IFNγ + TNFα/IL-10, while showing an almost two-fold relative risk of relapse in both cases, was not statistically significant. The limited sample size, the lack of healthy controls, and the fact that 94.1% of our patients were low expressers of TNFα might have influenced the strength of the inference. 

### 4.2. HLA Polymorphism

The HLA polymorphism within the studied population differs significantly from other populations, for example, the European population, with respect to the involvement of alleles not existing in the latter (Table 5).

The HLA polymorphism observed within the Ethiopian cohort with CML, when increased in number and compared to ‘healthy’ individuals, will form the basis for future analyses on the prevalence of the disease in Ethiopia. The latter is a limitation of the study presented here. We compared the results obtained from the favorable vs. the unfavorable group of patients as stated below. These results might offer the possibility to select patients for treatment on the basis of their immunogenetic background. It forms the basis for further testing. In this respect, extended typing via next- or third-generation sequencing seems to be the method of choice. 

In general, in this study, we observed some rare and new alleles, which need confirmation and submission in the IGMT/HLA database. A comparison with a control cohort was not within the scope of the present study; instead, we concentrated on the possible involvement of HLA and cytokines in the results of TKI treatment. Nevertheless, the results of the CML group reveal the following: HLA-*A*03:01:01* was the second most frequent HLA-A allele in the population of Ethiopian patients with CML and third in the total MHC class I alleles. HLA-A*03 was reported to be associated with a lower risk in the European cohort, when co-expressed with HLA-B8, which in Europe represents the allele HLA-*B*08:01*. In the Ethiopian cohort analyzed here, HLA-*B*08:01* was observed in about 2% of the two CML subgroups. With respect to the co-expression of the two alleles, we found two cases of co-expression versus two cases on non-co-expression. The HLA-DRB1*04 allele group was found in 12% of the Ethiopian patients with CML, far more than the 2% reported for an East African population by Peterson et al. [51]. We were aware that due to the low number of patients in both groups, these data must be confirmed in a larger cohort. A differential association was observed for *DRB4*01:01:01*, which demonstrated significant accumulation in the unfavorable group (9% vs. 2%), while the HLA allele *DRB4*01:03:01* accumulated in the favorable group (29% vs. 18%). Interestingly, the alleles of the DRB1*11 allele group were almost completely missing (2/162 found), confirming the results of Fort et al. [75]. These alleles have been reported to be associated with CML. All other alleles and allele groups were similarly expressed in both cohorts of favorable and unfavorable treatment outcomes, with the exception of the non-availability of the HLA-DRB345 genes. This is the case for the haplotypes DRB1*01, DRB1*08, and DRB1*10 (*p* = 0.0183). This warrants further clarification in a more extended study.

These preliminary explorative studies revealed the possible associations of the CML clinical outcome with alleles of HLA and cytokine SNPs. Besides the need for a standardized mechanism of associating SNPs with their function, multi-SNPs, surpassing single SNPs, and HLA allele polymorphisms showed promise in predicting patient outcomes during TKI treatment, prompting further exploration for future TFR trials.

## Figures and Tables

**Figure 1 genes-15-00732-f001:**
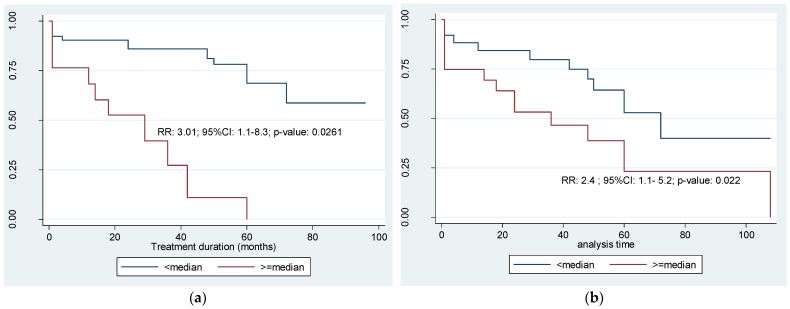
Age–adjusted survival function: (**a**) ratio of TNFα + IL-6 + TGFβ1/IL-10 + IFNγ; (**b**) ratio of IL-6/IL-10 + TGFβ1; (**c**) ratio of IL-6 + TNFα/IL-10; and (**d**) ratio of IFNγ + TNFα/IL-10.

**Table 1 genes-15-00732-t001:** Clinical features of patients with chronic myeloid leukemia.

Clinical Feature of Total CML Cohort	n	%
Number of patients	63	
Male-to-female ratio	2	58.7:41.3
Stage of disease at diagnosis		
CP	59	93.7
AP	2	3.2
BP	2	3.2
Splenomegaly at diagnosis	62	98
Hemoglobin at diagnosis		
</=8 g/dL *	10	15.9
<12 g/dL	35	55.5
>/=12 g/dL	18	28.6
Sokal risk score at diagnosis		
Low	3	4.8
Intermediate	16	25.4
High	44	69.8
WBC < 100,000/µL at diagnosis	15	23.8
Platelet > 450,000/µL at diagnosis	17	26.9
Median age at diagnosis in years (range)	37(19–69)	
Median treatment duration, months (range)	48 (1–144)	
Current BCR-ABL 1% (IS); median (95% CI)	33.8 (3.4–38.0)	

* According to the NCCN guidelines for cancer- and chemotherapy-induced anemia, a hemoglobin level of ≤8 g/dL is defined as severe-grade anemia. AC, accelerated phase; BC, blast crises; and CP, the clinical status of patients in the chronic phase of chronic myeloid leukemia.

**Table 2 genes-15-00732-t002:** Post-treatment clinical features of patients with CML.

Post-Treatment Clinical Features (n = 51 *)	Clinical Outcome, Median (Range) **	
	Favorable (n = 22)	Unfavorable (n = 29)
Treatment duration, months	60 (9–144)	36 (1–144)
WBC × 10^3^/µL	6.05 (3.4–10.7)	19.1 (2.8–180)
Platelets t × 10^3^/µL	225.5 (74–390)	182 (24–1314)
Hemoglobin, g/dL	14,2 (13–17)	12 (7.6–16.9)
BCR::ABL1 (IS)	0.02 (0.0001–0.25)	42.2 (2.02–78.22)

* Patients with BC, AP, and who were treatment naïve were excluded; ** clinical decision based on standard guidelines; international scale (IS); total white blood cell count (WBC).

**Table 3 genes-15-00732-t003:** Cytokine polymorphism in CML favorable and unfavorable groups.

Cytokine	Genotype	All Patients with CML (n = 63)		Only TKI Treated Patients with CML (n = 51) **				*p*-Value *
				Favorable Group (n = 22)		Unfavorable Group (n = 29)		
		n	%	n	%	n	%	
TNF-α (promoter −308G, −308A)	G/G low	60	95.2	21	95.5	27	93.1	1.000
	G/A high	2	3.2	1	4.5	1	3.4	1.000
	A/A high	1	1.2	0	0.0	1	3.4	1.000
IL-6 (promoter −174C, −174G)	C/C low	3	4.8	0	0.0	1	3.4	1.000
	G/G high	48	76.2	19	86.4	20	69.0	0.193
	G/C high	12	19.0	3	13.6	8	27.6	0.312
IFN-γ (intron +874T, +874A)	A/A low	31	49.2	9	40.9	18	62.1	0.164
	T/A intermediate	25	39.7	11	50.0	9	31.0	0.248
	T/T high	7	11.1	2	9.1	2	6.9	1.000
TGF-β (codon 10T, 10C, 25C, 25G)	C/C G/C low	2	3.2	2	9.1	0	0.0	0.181
	C/C C/C low	1	1.6	0	0.0	1	3.4	1.000
	T/T C/C low	0	0.0	0	0.0	0	0.0	na
	T/C C/C low	0	0.0	0	0.0	0	0.0	na
	T/C G/C intermediate	6	9.5	1	4.5	4	13.8	0.375
	C/C G/G intermediate	9	14.3	3	13.6	5	17.2	1.000
	T/T G/C intermediate	0	0.0	0	0.0	0	0.0	1.000
	T/T G/G high	22	34.9	7	31.8	11	37.9	0.770
	T/C G/G high	23	36.5	9	40.9	8	27.6	0.377
IL-10 (promoter −1082A, −1082G, −819T, −819C, −592A, −592C)	ACC/ACC low	1	1.6	0	0.0	0	0.0	na
	ACC/ATA low	11	17.5	5	22.7	6	20.7	1.000
	ATA/ATA low	3	4.8	1	4.5	2	6.9	1.000
	GCC/ACC intermediate	18	28.6	3	13.6	9	31.0	0.193
	GCC/ATA intermediate	16	25.4	5	22.7	6	20.7	1.000
	GCC/GCC high	14	22.2	8	36.4	6	20.7	0.342

* *p*-value shows there is no significant difference between favorable and unfavorable groups regarding the cytokine polymorphism. na = not applicable; ** only patients in the chronic phase were included.

**Table 4 genes-15-00732-t004:** HLA class I (A, B, C) and HLA class II (DRB1, DQB1, DQA1, DPA1, DPB1) polymorphisms in an Ethiopian population with CML.

**Locus**					**Favorable Group**		**Unfavorable Group**		
**HLA-A**	**Phenotype Count (n = 162)**	**Allele Count (2n = 324)**	**Phenotype Frequency (%)**	**Frequency (%)**	**Allele Count (2n = 96)**	**Allele Frequency (%)**	**Allele Count (2n = 206)**	**Allele Frequency (%)**	**Fisher Exact (*p*-Value)**
A*01:01:01	22.0	23.0	0.2	7.1	10	10.4	13	6.3	ns
A*01:02:01	1.0	1.0	0.0	0.3	1	1.0	0	0	ns
A*01:03:01	21.0	21.0	0.2	6.5	7	7.3	12	5.8	ns
A*02:01:01	50.0	52.0	0.4	16.0	13	13.5	37	18.0	ns
A*02:02:01	26.0	26.0	0.2	8.0	13	13.5	12	5.8	ns
A*02:05:01	12.0	12.0	0.1	3.7	5	5.2	7	3.4	ns
A*02:05:02	1.0	1.0	0.0	0.3	0	0	1	0.5	ns
A*02:06:01	1.0	1.0	0.0	0.3	0	0	1	0.5	ns
A*02:104	1.0	1.0	0.0	0.3	1	1.0	0	0	ns
A*02:14	1.0	1.0	0.0	0.3	0	0	1	0.5	ns
A*02:22:01	4.0	4.0	0.0	1.2	1	1.0	2	1.0	ns
**A*03:01:01**	28.0	29.0	0.2	9.0	5	5.2	23	11.2	**0.0347 ***
**A*03:02:01**	2.0	2.0	0.0	0.6	0	0	2	1.0	
A*11:01:01	3.0	3.0	0.0	0.9	2	2.1	1	0.5	ns
A*23:01:01	5.0	5.0	0.0	1.5	1	1.0	4	1.9	ns
**A*23:17:01**	10.0	10.0	0.1	3.1	0	0	9	4.4	**0.0285 ****
A*24:02:01	5.0	5.0	0.0	1.5	1	1.0	4	1.9	ns
A*26:01:01	1.0	1.0	0.0	0.3	0	0	0	0	ns
A*29:02:01	1.0	1.0	0.0	0.3	0	0	1	0.5	ns
A*30:01:01	19.0	19.0	0.2	5.9	2	2.1	14	6.8	ns
A*30:02:01	19.0	19.0	0.2	5.9	5	5.2	11	5.3	ns
A*30:04:01	7.0	7.0	0.1	2.2	2	2.1	5	2.4	ns
A*30:10	5.0	6.0	0.0	1.9	2	2.1	3	1.5	ns
A*31:04	1.0	1.0	0.0	0.3	0	0	1	0.5	ns
A*32:01:01	10.0	10.0	0.1	3.1	3	3.1	7	3.4	ns
A*33:03:01	4.0	4.0	0.0	1.2	0	0	4	1.9	ns
A*33:123N	2.0	2.0	0.0	0.6	0	0	2	1.0	ns
A*66:01:01	10.0	10.0	0.1	3.1	4	4.2	5	2.4	ns
A*66:03:01	3.0	3.0	0.0	0.9	0	0	3	1.5	ns
A*68:01:01	10.0	10.0	0.1	3.1	4	4.2	6	2.9	ns
A*68:02:01	26.0	27.0	0.2	8.3	8	8.3	14	6.8	ns
A*74:01:01	3.0	3.0	0.0	0.9	3	3.1	0	0	ns
A*74:03	4.0	4.0	0.0	1.2	3	3.1	1	0.5	ns
**Locus**					**Favorable group**		**Unfavorable group**		
**HLA-B**	**phenotype count (n = 162)**	**allele count (2n = 324)**	**phenotype frequency (%)**	**frequency (%)**	**allele count (2n = 96)**	**allele frequency (%)**	**allele count (2n = 206)**	**allele frequency (%)**	**Fisher exact (*p*-value)**
B*07:02:01	31.0	31.0	0.2	9.6	12	12.5	17	8.3	ns
B*07:05:01	15.0	16.0	0.1	4.9	3	3.1	10	4.9	ns
B*07:06:01	1.0	1.0	0.0	0.3	0	0	1	0.5	ns
B*08:01:01	6.0	6.0	0.0	1.9	2	2.1	4	1.9	ns
B*13:02:01	16.0	17.0	0.1	5.2	6	6.3	11	5.3	ns
B*14:01:01	2.0	2.0	0.0	0.6	1	1	1	0.5	ns
B*14:02:01	13.0	13.0	0.1	4.0	3	3.1	10	4.9	ns
B*15:10:01	5.0	5.0	0.0	1.5	1	1	3	1.5	ns
B*15:16:01	1.0	1.0	0.0	0.3	0	0	1	0.5	ns
B*15:17:01	4.0	4.0	0.0	1.2	1	1	3	1.5	ns
B*15:18:01	2.0	2.0	0.0	0.6	0	0	2	1	ns
B*15:220:01	24.0	25.0	0.2	7.7	7	7.3	17	8.3	ns
B*18:01:01	7.0	7.0	0.1	2.2	3	3.1	4	1.9	ns
B*27:03	6.0	6.0	0.0	1.9	2	2.1	4	1.9	ns
B*27:05:02	1.0	1.0	0.0	0.3	0	0	1	0.5	ns
B*27:26	1.0	1.0	0.0	0.3	0	0	1	0.5	ns
B*35:01:01	5.0	5.0	0.0	1.5	1	1	4	1.9	ns
B*35:03:01	1.0	1.0	0.0	0.3	0	0	1	0.5	ns
B*35:08:01	3.0	3.0	0.0	0.9	0	0	3	1.5	ns
B*37:01:01	1.0	1.0	0.0	0.3	1	1	0	0	ns
B*39:10:01	10.0	10.0	0.1	3.1	3	3.1	5	2.4	ns
B*39:24:01	4.0	5.0	0.0	1.5	2	2.1	3	1.5	ns
B*40:01:02	3.0	3.0	0.0	0.9	1	1	2	1	ns
B*40:02:01	1.0	1.0	0.0	0.3	0	0	1	0.5	ns
B*41:01:01	16.0	16.0	0.1	4.9	5	5.2	7	3.4	ns
B*41:02:01	6.0	6.0	0.0	1.9	3	3.1	2	1	ns
B*44:03:01	16.0	18.0	0.1	5.6	4	4.2	14	6.8	ns
B*44:03:02	1.0	1.0	0.0	0.3	0	0	1	0.5	ns
B*47:01:01	5.0	6.0	0.0	1.9	1	1	5	2.4	ns
B*49:01:01	25.0	25.0	0.2	7.7	3	3.1	19	9.2	ns
B*50:01:01	4.0	4.0	0.0	1.2	3	3.1	1	0.5	ns
B*51:01:01	12.0	13.0	0.1	4.0	3	3.1	9	4.4	ns
B*51:08:01	2.0	2.0	0.0	0.6	1	1	1	0.5	ns
B*53:01:01	24.0	24.0	0.2	7.4	8	8.3	15	7.3	ns
B*55:01:01	1.0	1.0	0.0	0.3	0	0	0	0	ns
**B*57:01:01**	3.0	3.0	0.0	0.9	1	1	2	1	**0.037 ***
**B*57:02:01**	5.0	5.0	0.0	1.5	2	2.1	2	1	
**B*57:03:01**	12.0	12.0	0.1	3.7	7	7.3	5	2.4	**0.0451 ****
B*58:01:01	18.0	18.0	0.1	5.6	5	5.2	12	5.8	ns
B*73:01	2.0	2.0	0.0	0.6	1	1	1	0.5	ns
B*81:02:02	1.0	1.0	0.0	0.3	0	0	1	0.5	ns
**Locus**					**Favorable group**		**Unfavorable group**		
**HLA-C**	**phenotype count (n = 162)**	**allele count (2n = 324)**	**phenotype frequency (%)**	**frequency (%)**	**allele count (2n = 96)**	**allele frequency (%)**	**allele count (2n = 206)**	**allele frequency (%)**	**Fisher exact (*p*-value)**
C*02:02:02	9.0	9.0	0.1	2.8	2	2.1	7	3.4	ns
C*03:02:01	1.0	1.0	0.0	0.3	1	1	0	0	ns
C*03:02:02	10.0	10.0	0.1	3.1	2	2.1	7	3.4	ns
C*03:04:01	2.0	2.0	0.0	0.6	0	0	2	1	ns
C*03:04:02	5.0	5.0	0.0	1.5	1	1	3	1.5	ns
C*04:01:01	43.0	46.0	0.3	14.2	12	12.5	31	15	ns
C*04:07:01	2.0	2.0	0.0	0.6	2	2.1	0	0	ns
C*06:02:01	40.0	43.0	0.3	13.3	17	17.7	26	12.6	ns
C*07:01:01	43.0	44.0	0.3	13.6	9	9.4	32	15.5	ns
C*07:01:02	10.0	10.0	0.1	3.1	5	5.2	5	2.4	ns
C*07:02:01	33.0	33.0	0.3	10.2	11	11.5	20	9.7	ns
C*07:04:01	8.0	8.0	0.1	2.5	4	4.2	4	1.9	ns
C*07:06:01	1.0	1.0	0.0	0.3	0	0	1	0.5	ns
C*07:18:01	7.0	7.0	0.1	2.2	2	2.1	5	2.4	ns
C*07:623	1.0	1.0	0.0	0.3	0	0	1	0.5	ns
C*08:02:01	13.0	13.0	0.1	4.0	3	3.1	10	4.9	ns
C*08:183	1.0	1.0	0.0	0.3	0	0	1	0.5	ns
C*08:28	1.0	1.0	0.0	0.3	1	1	0	0	ns
C*12:03:01	9.0	9.0	0.1	2.8	3	3.1	4	1.9	ns
C*12:167	1.0	1.0	0.0	0.3	0	0	1	0.5	ns
C*14:02:01	2.0	2.0	0.0	0.6	1	1	1	0.5	ns
C*14:03:01	2.0	2.0	0.0	0.6	0	0	2	1	ns
C*15:02:01	2.0	2.0	0.0	0.6	0	0	1	0.5	ns
C*15:05:01	5.0	5.0	0.0	1.5	2	2.1	3	1.5	ns
C*15:05:02	16.0	17.0	0.1	5.2	3	3.1	11	5.3	ns
C*16:01:01	5.0	5.0	0.0	1.5	3	3.1	2	1	ns
C*16:02:01	6.0	6.0	0.0	1.9	1	1	5	2.4	ns
C*16:04:01	6.0	6.0	0.0	1.9	1	1	5	2.4	ns
C*17:01:01	19.0	19.0	0.2	5.9	6	6.3	8	3.9	ns
C*17:03:01	4.0	4.0	0.0	1.2	2	2.1	2	1	ns
C*18:01:01	1.0	1.0	0.0	0.3	1	1	0	0	ns
C*18:02:01	8.0	8.0	0.1	2.5	1	1	6	2.9	ns
**Locus**					**Favorable group**		**Unfavorable group**		
**HLA-DRB1**	**phenotype count (n = 162)**	**allele count (2n = 324)**	**phenotype frequency (%)**	**frequency (%)**	**allele count (2n = 96)**	**allele frequency (%)**	**allele count (2n = 206)**	**allele frequency (%)**	**Fisher exact (*p*-value)**
DRB1*01:02:01	36	40	0.2	12.3	9	9.4	26	12.6	ns
DRB1*03:01:01	30	30	0.2	9.3	11	11.5	17	8.3	ns
DRB1*03:144	3	3	0.0	0.9	0	0.0	3	1.5	ns
DRB1*04:01:01	7	7	0.0	2.2	3	3.1	4	1.9	ns
DRB1*04:03:01	9	9	0.1	2.8	1	1.0	7	3.4	ns
DRB1*04:04:01	10	10	0.1	3.1	5	5.2	4	1.9	ns
DRB1*04:05:01	8	8	0.0	2.5	4	4.2	4	1.9	ns
DRB1*04:06:01	1	1	0.0	0.3	1	1.0	0	0.0	ns
DRB1*04:06:02	2	2	0.0	0.6	0	0.0	2	1.0	ns
DRB1*04:08:01	1	1	0.0	0.3	0	0.0	1	0.5	ns
DRB1*04:201	1	1	0.0	0.3	0	0.0	1	0.5	ns
DRB1*07:01:01	57	64	0.4	19.8	17	17.7	40	19.4	ns
DRB1*08:04:01	16	16	0.1	4.9	4	4.2	10	4.9	ns
DRB1*08:08	4	4	0.0	1.2	1	1.0	3	1.5	ns
DRB1*10:01:01	8	8	0.0	2.5	2	2.1	6	2.9	ns
DRB1*11:01:01	5	5	0.0	1.5	0	0.0	4	1.9	ns
DRB1*11:01:02	1	1	0.0	0.3	0	0.0	1	0.5	ns
DRB1*13:01:01	6	6	0.0	1.9	2	2.1	4	1.9	ns
DRB1*13:02:01	56	63	0.3	19.4	20	20.8	43	20.9	ns
DRB1*13:03:01	11	12	0.1	3.7	5	5.2	7	3.4	ns
DRB1*14:01:01	1	1	0.0	0.3	0	0.0	1	0.5	ns
DRB1*14:54:01	5	5	0.0	1.5	3	3.1	1	0.5	ns
DRB1*15:01:01	3	3	0.0	0.9	1	1.0	2	1.0	ns
DRB1*15:03:01	24	24	0.1	7.4	7	7.3	15	7.3	ns
**HLA-DRB345**	**phenotype count (n = 162)**	**allele count (2n = 324)**	**phenotype frequency (%)**	**frequency (%)**	**allele count (2n = 96)**	**allele frequency (%)**	**allele count (2n = 206)**	**allele frequency (%)**	**Fisher exact (*p*-value)**
DRB3*01:01:02	18	21	0.1	6.5	5	5.2	15	7.3	ns
DRB3*02:02:01	47	53	0.3	16.4	18	18.8	30	14.6	ns
DRB3*03:01:01	54	70	0.3	21.6	19	19.8	51	24.8	ns
DRB3*03:15	1	1	0.0	0.3	1	1.0	0	0.0	ns
**DRB4*01:01:01**	13	21	0.1	6.5	2	2.1	19	9.2	**0.0127**
**DRB4*01:03:01**	69	77	0.4	23.8	28	29.2	36	17.5	**0.0058**
DRB4*01:03:01:02N	15	16	0.1	4.9	4	4.2	12	5.8	ns
DRB4*01:03:03	6	6	0.0	1.9	2	2.1	4	1.9	ns
DRB5*01:01:01	30	31	0.2	9.6	9	9.4	20	9.7	ns
non #	28	28	0.2	8.6	8	8.3	19	9.2	ns
**Locus**					**Favorable group**		**Unfavorable group**		
**HLA-DQA1**	**phenotype count (n = 162)**	**allele count (2n = 324)**	**phenotype frequency (%)**	**frequency (%)**	**allele count (2n = 96)**	**allele frequency (%)**	**allele count (2n = 206)**	**allele frequency (%)**	**Fisher exact (*p*-value)**
DQA1*01:01:02	36	40	0.2	12.3	9	9.4	26	12.6	ns
DQA1*01:02:01	75	82	0.5	25.3	26	27.1	55	26.7	ns
DQA1*01:02:02	8	8	0.0	2.5	4	4.2	3	1.5	ns
DQA1*01:03:01	6	6	0.0	1.9	2	2.1	4	1.9	ns
DQA1*01:04:01	3	3	0.0	0.9	2	2.1	0	0.0	ns
DQA1*01:05:01	8	8	0.0	2.5	2	2.1	6	2.9	ns
DQA1*01:05:02	3	3	0.0	0.9	1	1.0	2	1.0	ns
DQA1*02:01:01	57	64	0.4	19.8	15	15.6	42	20.4	ns
DQA1*03:01:01	14	14	0.1	4.3	3	3.1	10	4.9	ns
DQA1*03:03:01	25	25	0.2	7.7	11	11.5	13	6.3	ns
DQA1*04:01:01	4	4	0.0	1.2	1	1.0	3	1.5	ns
DQA1*04:01:02	4	4	0.0	1.2	2	2.1	2	1.0	ns
DQA1*05:01:01	32	33	0.2	10.2	11	11.5	20	9.7	ns
DQA1*05:05:01	30	30	0.2	9.3	7	7.3	20	9.7	ns
**HLA-DQB1**	**phenotype count (n = 162)**	**allele count (2n = 324)**	**phenotype frequency (%)**	**frequency (%)**	**allele count (2n = 96)**	**allele frequency (%)**	**allele count (2n = 206)**	**allele frequency (%)**	**Fisher exact (*p*-value)**
DQB1*02:01:01	31	33	0.2	10.2	12	12.5	19	9.2	ns
DQB1*02:02:01	56	64	0.3	19.8	16	16.7	41	19.9	ns
DQB1*03:01:01	32	33	0.2	10.2	9	9.4	22	10.7	ns
DQB1*03:01:04	3	3	0.0	0.9	0	0.0	2	1.0	ns
DQB1*03:02:01	16	16	0.1	4.9	5	5.2	11	5.3	ns
DQB1*03:03:02	7	7	0.0	2.2	3	3.1	4	1.9	ns
DQB1*04:02:01	10	10	0.1	3.1	5	5.2	4	1.9	ns
DQB1*04:02:13	2	2	0.0	0.6	1	1.0	1	0.5	ns
DQB1*04:87	4	4	0.0	1.2	1	1.0	3	1.5	ns
DQB1*05:01:01	44	49	0.3	15.1	11	11.5	32	15.5	ns
DQB1*05:03:01	3	3	0.0	0.9	2	2.1	0	0.0	ns
DQB1*06:02:01	23	23	0.1	7.1	5	5.2	17	8.3	ns
DQB1*06:03:01	11	11	0.1	3.4	4	4.2	6	2.9	ns
DQB1*06:04:01	41	45	0.3	13.9	16	16.7	29	14.1	ns
DQB1*06:09:01	20	20	0.1	6.2	5	5.2	15	7.3	ns
DQB1*06:27:01	1	1	0.0	0.3	1	1.0	0	0.0	ns
**Locus**					**Favorable group**		**Unfavorable group**		
**HLA-DPA1**	**phenotype count (n = 162)**	**allele count (2n = 324)**	**phenotype frequency (%)**	**frequency (%)**	**allele count (2n = 96)**	**allele frequency (%)**	**allele count (2n = 206)**	**allele frequency (%)**	**Fisher exact (*p*-value)**
DPA1*01:03:01	163.0	184.0	1.0	56.8	55	57.3	118	57.3	ns
DPA1*01:04	6.0	6.0	0.0	1.9	2	2.1	4	1.9	ns
DPA1*01:04:01	18.0	20.0	0.1	6.2	4	4.2	15	7.3	ns
DPA1*01:05	1.0	1.0	0.0	0.3	0	0.0	1	0.5	ns
DPA1*01:30	5.0	5.0	0.0	1.5	1	1.0	4	1.9	ns
DPA1*01:58	5.0	5.0	0.0	1.5	2	2.1	3	1.5	ns
DPA1*01:58:01	1.0	1.0	0.0	0.3	0	0.0	1	0.5	ns
DPA1*02:01:01	73.0	79.0	0.5	24.4	23	24.0	48	23.3	ns
DPA1*02:01:07	4.0	4.0	0.0	1.2	1	1.0	2	1.0	ns
DPA1*02:02:02	5.0	5.0	0.0	1.5	4	4.2	1	0.5	ns
DPA1*02:09	4.0	4.0	0.0	1.2	1	1.0	2	1.0	ns
DPA1*03:01:01	8.0	8.0	0.0	2.5	2	2.1	6	2.9	ns
DPA1*03:01:02	1.0	1.0	0.0	0.3	0	0.0	1	0.5	ns
DPA1*04:02	1.0	1.0	0.0	0.3	1	1.0	0	0.0	ns
**HLA-DPB1**	**phenotype count (n = 162)**	**allele count (2n = 324)**	**phenotype frequency (%)**	**frequency (%)**	**allele count (2n = 96)**	**allele frequency (%)**	**allele count (2n = 206)**	**allele frequency (%)**	**Fisher exact (*p*-value)**
DPB1*01:01:01	2.0	2.0	0.0	0.6	3	3.1	1	0.5	ns
DPB1*01:01:02	5.0	5.0	0.0	1.5	0	0.0	2	1.0	ns
DPB1*02:01:02	71.0	76.0	0.4	23.5	23	24.0	50	24.3	ns
DPB1*02:01:18	1.0	1.0	0.0	0.3	0	0.0	1	0.5	ns
DPB1*03:01:01	12.0	12.0	0.1	3.7	2	2.1	9	4.4	ns
DPB1*04:01:01	63.0	65.0	0.4	20.1	19	19.8	43	20.9	ns
DPB1*04:02:01	5.0	5.0	0.0	1.5	1	1.0	3	1.5	ns
DPB1*05:01:01	1.0	1.0	0.0	0.3	1	1.0	0	0.0	ns
DPB1*09:01:01	6.0	6.0	0.0	1.9	0	0.0	6	2.9	ns
DPB1*104:01:01	15.0	16.0	0.1	4.9	7	7.3	7	3.4	ns
DPB1*105:01:01	2.0	2.0	0.0	0.6	0	0.0	2	1.0	ns
DPB1*105:02:01	1.0	1.0	0.0	0.3	0	0.0	1	0.5	ns
DPB1*11:01:01	17.0	18.0	0.1	5.6	6	6.3	8	3.9	ns
DPB1*124:01:01	3.0	3.0	0.0	0.9	1	1.0	2	1.0	ns
DPB1*13:01:01	12.0	13.0	0.1	4.0	4	4.2	6	2.9	ns
DPB1*133:01	1.0	1.0	0.0	0.3	0	0.0	1	0.5	ns
DPB1*135:01:01	1.0	1.0	0.0	0.3	1	1.0	0	0.0	ns
DPB1*14:01:01	6.0	6.0	0.0	1.9	3	3.1	3	1.5	ns
DPB1*15:01:01	27.0	29.0	0.2	9.0	8	8.3	19	9.2	ns
DPB1*17:01:01	33.0	34.0	0.2	10.5	8	8.3	24	11.7	ns
DPB1*290:01	1.0	1.0	0.0	0.3	0	0.0	1	0.5	ns
DPB1*30:01:01	7.0	7.0	0.0	2.2	2	2.1	5	2.4	ns
DPB1*34:01:01	4.0	4.0	0.0	1.2	2	2.1	2	1.0	ns
DPB1*39:01:01	4.0	4.0	0.0	1.2	0	0.0	4	1.9	ns
DPB1*47:01:01	2.0	2.0	0.0	0.6	1	1.0	1	0.5	ns
DPB1*49:01:01	1.0	1.0	0.0	0.3	0	0.0	1	0.5	ns
DPB1*54:01	2.0	2.0	0.0	0.6	0	0.0	2	1.0	ns
DPB1*55:01:01	4.0	4.0	0.0	1.2	2	2.1	2	1.0	ns
DPB1*665:01:01	1.0	1.0	0.0	0.3	1	1.0	0	0.0	ns
DPB1*702:01	1.0	1.0	0.0	0.3	1	1.0	0	0.0	ns

** p*-value calculated at the antigen level; ** *p*-value calculated at the allele level; # characterizes individuals with missing *DRB3*, *DRB4*, and *DRB5* genes; ns indicate non-significant results; bold indicates significant results.

## Data Availability

The data supporting this study’s findings are available from the corresponding author upon reasonable request.

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
