# Peer review of "Prognostic Role of Human Leukocyte Antigen Alleles and Cytokine Single-Nucleotide Polymorphisms in Patients with Chronic Myeloid Leukemia Treated with Tyrosine Kinase Inhibitor Drugs"

_genes, 2024, doi:10.3390/genes15060732_

Round 1
Reviewer 1 Report (Previous Reviewer 2)
Comments and Suggestions for Authors
Thank you very much for submitting your manuscript to the journal "genes" and for your great revision compared to the version that you have submitted in 2023. Attached you will find some minor/major concerns regarding your manuscript:
- The manuscript lacks rigorous and clarity in many places. Much of the text could be replaced by simple sentences to improve the flow of speech.
- Line 549: Please clarify "enhanced effects". The 2-3 summarising sentences could be expressed more precisely.
- Figure 1 could be mentioned/discussed more
- Extensive linguistic revision (Commas, punctuation, and spelling mistakes). A few examples: "patients' immune" (graphical abstract), line 417 "a", line 435 "lacks", line 436 remove "the", too many spaces at many locations, line 539-540 "CML" is already introduced.
- Please discuss a shortening of the introduction.
- Abbreviations in the main text should be introduced (e.g. LSC)
Comments on the Quality of English Language
A thorough revision (especially due to spelling mistakes) is necessary.
Author Response
Thanks to the reviewer for the comments and findings.
Thank you very much for submitting your manuscript to the journal "genes" and for your great revision compared to the version that you have submitted in 2023. Attached you will find some minor/major concerns regarding your manuscript:
- The manuscript lacks rigorous and clarity in many places. Much of the text could be replaced by simple sentences to improve the flow of speech. The manuscript has been revised and shortened in appropriate places, see track change
- Line 549: Please clarify "enhanced effects". The 2-3 summarising sentences could be expressed more precisely. Done
- Figure 1 could be mentioned/discussed more This the paragraph has been changed and corrected
- Extensive linguistic revision (Commas, punctuation, and spelling mistakes).
A few examples:
"patients' immune" (graphical abstract), corrected
line 417 "a", corrected
line 435 "lacks", corrected
line 436 remove "the", corrected
too many spaces at many locations, corrected
line 539-540 "CML" is already introduced. This the paragraph has been changed and corrected
- Please discuss a shortening of the introduction. done
- Abbreviations in the main text should be introduced (e.g. LSC)
corrected
Comments on the Quality of English Language
A thorough revision (especially due to spelling mistakes) is necessary.
We will use the service of MDPI for the English language revision
Reviewer 2 Report (Previous Reviewer 1)
Comments and Suggestions for Authors
Although I am concerned about the limited number of patients analyzed in this paper, I acknowledge that the authors have answered all of my comments in a detailed and thorough manner, which has significantly improved the paper.
Author Response
Although I am concerned about the limited number of patients analyzed in this paper, I acknowledge that the authors have answered all of my comments in a detailed and thorough manner, which has significantly improved the paper.
Thanks you so much for your efforts
Round 2
Reviewer 1 Report (Previous Reviewer 2)
Comments and Suggestions for Authors
Thank you very much for the (linguistic) revision of your manuscript.
Comments on the Quality of English Language
Improved
This manuscript is a resubmission of an earlier submission. The following is a list of the peer review reports and author responses from that submission.
Round 1
Reviewer 1 Report
Comments and Suggestions for Authors
This paper is conceptually interesting and analyzes the role of inflammatory cytokines and HLA polymorphisms in responsiveness to TKI treatment in CML patients.
1. The small number of 63 cases analyzed and the inclusion of various stages of CML such as AP and BP seem to obscure this analysis. In order to increase the homogeneity of the cohort, the analyzed cases may be restricted to CP cases.
2. CML-CP rarely present with anemia at onset. The authors should explain whether the anemia in Table 1 is post-TKI treatment, an effect of AP/BP, or due to other causes such as secondary myelofibrosis.
3. How was treatment responsiveness, favorable vs. unfavorable, determined?
4. Additional chromosomal abnormalities are factors influencing TKI treatment responsiveness. Age and gender may have influenced the results of this study. Performing multivariate analysis may facilitate interpretation of the results.
5. The authors should clarify the functional role of the significantly affected genetic polymorphisms, especially their impact on TKI treatment responsiveness.
Author Response
This paper is conceptually interesting and analyzes the role of inflammatory cytokines and HLA polymorphisms in responsiveness to TKI treatment in CML patients.
- The small number of 63 cases analyzed and the inclusion of various stages of CML such as AP and BP seem to obscure this analysis. In order to increase the homogeneity of the cohort, the analyzed cases may be restricted to CP cases.
Accepted, we have included only Chronic phase patients for the cytokine analysis. But, we have included all for HLA analysis as the sample size is large enough not to be confounded by few of the blast or accelerated phase patients.
- CML-CP rarely present with anemia at onset. The authors should explain whether the anemia in Table 1 is post-TKI treatment, an effect of AP/BP, or due to other causes such as secondary myelofibrosis.
Thank you very much for the valuable comment, we have reviewed clinical records and indicated clinical data including anemia before and post TKI. But, anemia due to secondary myelofibrosis is not confirmed in any of these anemic patients.
- How was treatment responsiveness, favorable vs. unfavorable, determined?
Thank you very much for this good comment, it was according to the Chronic Myeloid Leukemia, Version 2.2021, NCCN . we have included as reference.
- Additional chromosomal abnormalities are factors influencing TKI treatment responsiveness. Age and gender may have influenced the results of this study. Performing multivariate analysis may facilitate interpretation of the results.
Thank you for comment, yes, we have data on age and gender, we have included in the overall clinical data of the patients. But, the sample size is too small for multivariate analysis, specially for the SNP of the cytokine gene.
- The authors should clarify the functional role of the significantly affected genetic polymorphisms, especially their impact on TKI treatment responsiveness.
Thank you for the comment. We elaborated this topic in the discussion of this re-submission in more detail.
Reviewer 2 Report
Comments and Suggestions for Authors
Your manuscript is about the impact of cytokines and HLA in patients with chronic myeloid leukemia treated at Addis Ababa University (Ethiopia). Your research is of great interest to better understand worse disease outcome despite therapy with TKIs. Attached you will find some major and minor issues regarding the manuscript.
Major Issue:
- 1. Line 96-97: The study was approved by the Institutional Review Board (IRB). Please provide the project-specific approval number.
- 2. Line 102: In how many cases treatment was discontinued or changed?
- 3. Linguistic revision of the manuscript is highly recommended.
- 4. Line 199: Please mention the groups, respectively.
- 5. Introduction: Please explain in more detail the gene polymorphisms you are focusing on.
- 6. Please discuss your results using a healthy comparison cohort.
- 7. Please discuss other confounding factors, such as WBCs and cell subtypes, patient ethnicity/race, gender and others.
- 8. Did any patient receive bone marrow tranplantation?
- 9. Please mention p-values/statistics in the main text.
- 10. Please discuss potential hypotheses regarding your observed findings. Is it possible to test new mechanisms ex vivo that may worsen the disease course.
- 11. Please discuss the relevance of the new alleles detected (line 300).
- 12. A graphical abstract/table summarizing your results could be helpful.
- 13. A table comparing your findings with the findings of other studies/other populations could be helpful.
Minor Issues:
- 1. Please introduce all abbreviations mentioned in your manuscript
- 2. Line 91: or “further classified”
- 3. Line 98: “was”
- 4. Line 105: Please discuss if you want to use “had a high Sokal Risk Score”
- 5. Line 112: “all patients received”
- 6. Line 231: Please discuss if you want to use “unpublished data” or “work in progress”
- 7. Line 267: “show”
Comments on the Quality of English Language
A linguistic revision is necessary.
Author Response
Your manuscript is about the impact of cytokines and HLA in patients with chronic myeloid leukemia treated at Addis Ababa University (Ethiopia). Your research is of great interest to better understand worse disease outcome despite therapy with TKIs. Attached you will find some major and minor issues regarding the manuscript.
Major Issue:
- 1. Line 96-97: The study was approved by the Institutional Review Board (IRB). Please provide the project-specific approval number.
Thank you very much for the comment. We included the project-specific approval number in line 169.
- 2. Line 102: In how many cases treatment was discontinued or changed?
The number of the patients included in the study for the study are given in table 2 of the revised manuscript.
- 3. Linguistic revision of the manuscript is highly recommended.
Thank you for this recommendation. We we have revised the language of the manuscript
- 4. Line 199: Please mention the groups, respectively.
Thank you for the comment. The groups are given in table 2 of the revised manuscript.
- 5. Introduction: Please explain in more detail the gene polymorphisms you are focusing on.
Thank you very much for the comment. It is done in the revised manuscript.
- 6. Please discuss your results using a healthy comparison cohort.
Unfortunately this is not possible in the present cohort. This current study is a preliminary explorative study. We plan to investigate, depending on the resources, a healthy control group from the same region of the patients. Therefore we only compared the results within the favorable and unfavorable group, knowing that is a strong limitation of the study.
- 7. Please discuss other confounding factors, such as WBCs and cell subtypes, patient ethnicity/race, gender and others.
Thank you for the comment. We included this point in the result in the Kaplan-Meier estimates.
- 8. Did any patient receive bone marrow tranplantation?
Non
- 9. Please mention p-values/statistics in the main text.
Done
- 10. Please discuss potential hypotheses regarding your observed findings. Is it possible to test new mechanisms ex vivo that may worsen the disease course.
In the moment we are not able to establish any laboratory assay to test such possibilities. This might be one of the focuses in the future.
- 11. Please discuss the relevance of the new alleles detected (line 300).
This is still a matter of investigation and in our opinion it is still too early to discuss any relevance.
- 12. A graphical abstract/table summarizing your results could be helpful.
Thank you very much for this idea. We added a graphic abstract.
- 13. A table comparing your findings with the findings of other studies/other populations could be helpful.
Thank you very much for this idea. We included table 5 in the revised manuscript.
Minor Issues:
- 1. Please introduce all abbreviations mentioned in your manuscript
Done in the revised manuscript.
- 2. Line 91: or “further classified”
This line is rephrased.
- 3. Line 98: “was”
This line is rephrased.
- 4. Line 105: Please discuss if you want to use “had a high Sokal Risk Score”
The clinical feature of the cohort are depicted in table 1. We used a diagnosis the Sokal risk score as parameter.
- 5. Line 112: “all patients received”
corrected